# Vitamin D Metabolite Ratio in Pregnant Women with Low Blood Vitamin D Concentrations Is Associated with Neonatal Anthropometric Data

**DOI:** 10.3390/nu14112201

**Published:** 2022-05-25

**Authors:** Tomozumi Takatani, Yuzuka Kunii, Mamoru Satoh, Akifumi Eguchi, Midori Yamamoto, Kenichi Sakurai, Rieko Takatani, Fumio Nomura, Naoki Shimojo, Chisato Mori

**Affiliations:** 1Department of Pediatrics, Graduate School of Medicine, Chiba University, Chiba 260-8670, Japan; yuzu.mamefii@gmail.com; 2Division of Clinical Mass Spectrometry, Chiba University Hospital, Chiba 260-8677, Japan; msatoh1995@yahoo.co.jp (M.S.); fnomura@faculty.chiba-u.jp (F.N.); 3Center for Preventive Medical Sciences, Chiba University, Chiba 263-8522, Japan; a_eguchi@chiba-u.jp (A.E.); midoriy@faculty.chiba-u.jp (M.Y.); sakuraik@faculty.chiba-u.jp (K.S.); rieko-takatani@chiba-u.jp (R.T.); shimojo@faculty.chiba-u.jp (N.S.); cmori@faculty.chiba-u.jp (C.M.); 4Department of Bioenvironmental Medicine, Graduate School of Medicine, Chiba University, Chiba 260-8670, Japan

**Keywords:** 25-hydroxyvitamin D, birth anthropometric data, 3-epi-25-hydroxyvitamin D_3_, vitamin D metabolite ratio, vitamin D insufficiency

## Abstract

Existing evidence on the correlation between maternal vitamin D concentrations and birth outcomes is conflicting. Investigation of these associations requires accurate assessment of vitamin D status, especially in individuals with low 25-hydroxyvitamin D (25(OH)D) concentrations. This study examined the correlations between birth outcomes and the maternal vitamin D metabolite ratio (VMR) 1 (defined as the ratio of 24,25(OH)_2_D_3_ to 25(OH)D) and VMR2 (defined as the ratio of 3-epi-25(OH)D_3_ to 25(OH)D) using data from the Japan Environment and Children’s Study at Chiba Regional Center. A total of 297 mother–neonate pairs were analyzed. Using liquid chromatography–tandem mass spectrometry, we measured 25(OH)D_2_, 25(OH)D_3_, 24,25(OH)_2_D_3_, and 3-epi-25(OH)D_3_ concentrations in maternal serum samples. These data were analyzed in relation to birth anthropometric data using multivariable linear regression. Of the study participants, 85.2% showed insufficient vitamin D concentrations. VMR1 was strongly correlated with 25(OH)D concentrations, whereas VMR2 showed a weak correlation. Only VMR2 was associated with all anthropometric data. VMR2 in pregnant women with low vitamin D blood concentrations is a useful marker for neonatal anthropometric data and is independent of 25(OH)D. Accurate measurement of vitamin D metabolites could help better understand the effects of vitamin D on birth outcomes.

## 1. Introduction

Intrauterine growth is regulated by multiple genetic and environmental factors. Although genetic factors have been proven to play a substantial role in intrauterine growth, accumulating evidence shows that the intrauterine environment contributes significantly more to fetal growth. A study using embryo transfer revealed that the recipient mother affected intrauterine growth more than the donor mother [1]. Other studies have suggested that 25–40% of the variability in birth weight can be explained by fetal genes [2,3,4]. Prenatal growth is an important public health concern worldwide because intrauterine growth restriction is a major cause of mortality in neonates and infants [5]. Moreover, prenatal growth is significantly associated with adult morbidity throughout life, such as cardiovascular disease, neurodevelopmental complications, and endocrine disorders [6,7,8]. In Japan, the average weight of a newborn has decreased over the last 3 decades from 3200 g to 3020 g [9]. Although several attempts have been made to determine which factors contribute to birth size, these factors are not fully known. Vitamin D is an important nutritional factor, and its deficiency is related to several negative health outcomes. However, scientific evidence for the association between vitamin D and birth outcomes is controversial [10,11,12,13,14,15,16,17,18,19].

A World Health Organization scientific group meeting defined Vitamin D insufficiency as total serum 25-hydroxyvitamin D (25(OH)D) concentrations of less than 20 ng/mL [20]. However, some ethnic groups who have low serum 25(OH)D concentrations show healthy bone mineral density, which is a major health outcome associated with 25(OH)D. African–Americans are representative of such groups with low 25(OH)D concentrations [21,22,23] and paradoxically show a higher bone mineral density and a lower risk of fracture than White Americans [24,25,26,27]. This finding suggests that the current measurement of 25(OH)D concentrations is not directly correlated with some health outcomes in certain groups. A more detailed analysis of vitamin D status may be required to determine its relationship to health outcomes.

A recent report has shown that the ratio of 24,25-dihydroxyvitamin D_3_ (24,25(OH)_2_D_3_) to 25(OH)D is a better risk indicator for hip fracture than 25(OH)D because this ratio reflects vitamin D receptor (VDR) activity [28]. Moreover, other reports have suggested that the C-3 epimer 25-hydroxyvitamin D_3_ (3-epi-25(OH)D_3_), which is converted from 25(OH)D_3_ by 3-epimerase in the liver, is also related to several health outcomes [29,30]. The availability of newer technologies, such as liquid chromatography–tandem mass spectrometry (LC-MS/MS), which can distinguish 3-epi-25(OH)D_3_ from 25(OH)D_3_, have increased interest in 3-epi-25(OH)D_3_ when considering its association with health outcomes. The metabolite 3-epi-25(OH)D_3_ is thought to be less potent physiologically than 25(OH)D, and the ratio of 3-epi-25(OH)D_3_ to 25(OH)D is associated with diabetic complications [31]. Both ratios can reflect vitamin D metabolism more closely than either metabolite alone, and both can be markers of several health outcomes independent of 25(OH)D. However, the associations between these ratios in pregnant women with low 25(OH)D concentrations and prenatal growth have not been investigated.

To address this lack of knowledge, we measured the ratio of 24,25(OH)_2_D_3_ to 25(OH)D, which we defined as the vitamin D metabolite ratio (VMR) 1, and the ratio of 3-epi-25(OH)D_3_ to 25(OH)D, which we defined as VMR2. To verify whether VMR1 and VMR2 are independent of 25(OH)D, we determined the correlations between 25(OH)D and VMR1 and between 25(OH)D and VMR2 by a Spearman correlation test. We analyzed their associations with birth outcomes using data from the Japan Environment and Children’s Study (JECS) at the Chiba Regional Center. The JECS is an ongoing birth cohort study. A report using JECS data has shown that most Japanese pregnant women had insufficient vitamin D concentrations [32,33]. Therefore, in this Japanese cohort, we clarified the associations between vitamin D metabolite concentrations in pregnant women who have low 25(OH)D concentrations and neonatal anthropometric data.

## 2. Materials and Methods

### 2.1. Study Participants

This study used data obtained from the Chiba Regional Center. This study was conducted as an adjunct study of the JECS funded by the Ministry of the Environment, Japan. The JECS protocol was described previously [34]. In 15 regional centers, the JECS recruited pregnant women in early pregnancy between January 2011 and March 2014. We analyzed jecs-ta-20190930 (released in October 2019), which is a dataset from the JECS. The JECS protocol was approved by the Institutional Review Board on Epidemiological Studies of the Ministry of the Environment and by the Ethics Committees of all participating institutions. The project code of this adjunct study at the Chiba Regional Center was #1538. Written informed consent was obtained from all participating women according to the Declaration of Helsinki.

### 2.2. Measurement of Vitamin D Metabolites

Of the 103,060 JECS pregnancies, 1965 mother-infant pairs with available maternal serum samples at Chiba Regional Center were included in this study. Maternal serum samples were collected before 22 weeks of pregnancy and between 22 weeks and 35 weeks of pregnancy. We randomly selected 400 maternal serum samples collected between 22 weeks and 35 weeks of pregnancy. Sampling was conducted throughout the year, and samples were stored at −80 °C until analysis. Vitamin D and its metabolites, such as 25(OH)D_2_, 25(OH)D_3_, 24,25(OH)_2_D_3_, and 3-epi-25(OH)D_3_, were measured using LC-MS/MS as previously described [35]. We are part of the Vitamin D External Quality Assessment Scheme (DEQAS). We calculated 25(OH)D as the sum of 25(OH)D_2_ and 25(OH)D_3_.

The standard reference material (SRM) 972a from the National Institute of Standards and Technology was used to examine the accuracy of the LC-MS/MS method in this study. The SRM 972a consists of four levels: 25(OH)D_2_, 25(OH)D_3_, 24,25(OH)_2_D_3_, and 3-epi-25(OH)D_3_ [36].

### 2.3. Covariate and Birth Outcome Data

We collected data on birth outcomes and covariates as previously described [34]. Anthropometric measurements at birth, the neonate’s sex, gestational period, maternal age at delivery, parity, and the mode of delivery were acquired from medical records transcribed during pregnancy and after delivery. Data on annual family income, education levels, and smoking and drinking habits during pregnancy were collected using self-administered questionnaires, which were filled in during registration to this study and during mid–late pregnancy. Maternal anthropometric data, maternal age, pre-pregnancy height, and weight were obtained from medical records and questionnaires. Maternal body mass index (BMI) was calculated as follows: body weight(kg)/(height(m))^2^.

### 2.4. Statistical Analysis

Simple and multiple linear regression models were used to calculate unadjusted and adjusted differences (β) and 95% confidence intervals (CIs) for weight, length, head circumference, and chest circumference at birth. Linear regression models were used to calculate expected differences in the birth anthropometric measurements for each 1 ng/mL increase in vitamin D metabolites or 1-unit increase in VMR1 or VMR2.Multiple linear regression models were adjusted for maternal age (<25, 25–29, 30–34, and ≥35 years), parity (0 and ≥1), pre-pregnancy BMI (<18.5, 18.5–< 25, and ≥25.0 kg/m^2^), maternal education (<10, 10–12, 13–16, and ≥17 years of education), household income (<2, 2 to <4, 4 to <6, 6 to <8, 8 to <10, and ≥10 million Japanese yen (JPY)), smoking habit during pregnancy (never smoked, ex-smokers who quit before pregnancy, and smoked during pregnancy), alcohol habit during pregnancy (never drank, ex-drinkers who quit before pregnancy, and drank during pregnancy), the mode of delivery (vaginal delivery, cesarean section), and the neonate’s sex. Covariates were selected a priori based on the literature and biological plausibility [37,38]. The Spearman correlation test was performed to determine the correlations between 25(OH)D and VMR1 and between 25(OH)D and VMR2.

For all analyses, *p*-values of less than 0.05 were used to denote statistical significance. All statistical analyses were performed using R (version 4.0.3; R Development Core Team, Vienna, Austria; http://www.R-project.org, accessed on 8 November 2020).

## 3. Results

After excluding mothers with stillbirth, miscarriage, and multiple births, 358 mother–neonate pairs were included in this study. Then, we excluded mother–neonate pairs with one or more missing data (i.e., birth weight, birth length, birth head circumference, birth chest circumference, age, BMI before pregnancy, parity, smoking during pregnancy, drinking during pregnancy, education level, annual household income, and mode of delivery). Finally, the completed dataset comprised 297 mother–neonate pairs (Figure 1).

### 3.1. Baseline Characteristics

The characteristics of the mothers and neonates enrolled in this study are shown in Table 1 and Table 2, respectively. The mean age of the mothers at delivery was 31.3 ± 4.8 years. Approximately 90% of the participants had a BMI of less than 25 kg/m^2^ before pregnancy. Moreover, 32.7% of the participants showed an annual household income between 2 and 4 million JPY, which is almost the median annual income of the Japanese working population. Most participants had more than 10 years of education, which represented high school or higher education. Overall, 53.5% of the mothers never smoked, whereas 15.2% smoked during early pregnancy. Additionally, 43.4% of the mothers drank during early pregnancy. The mean gestational age was 39.2 ± 1.5 weeks for 297 singletons.

### 3.2. Vitamin D Metabolite Concentrations

The mean, interquartile range, minimum, and maximum values of vitamin D metabolites, VMR1, and VMR2 are shown in Table 3. All metabolites were detected in all samples. Moreover, 85.2% of the participants had insufficient vitamin D concentrations when the threshold was set at 20 ng/mL (Table 1). The Spearman correlation test showed a strong correlation between VMR1 and 25(OH)D concentrations (r = 0.732; *p* < 0.001), whereas a weak correlation between VMR2 and 25(OH)D concentrations was found (r = 0.304; *p* < 0.001) (Figure 2). Quality control analyses of the vitamin D measurements, shown in Appendix A, demonstrated acceptable quality.

### 3.3. Factors including VMR and Neonatal Anthropometric Data

Unadjusted linear regression revealed that neither the individual vitamin D metabolites nor VMR1 were significantly associated with birth outcomes. Notably, VMR2 showed significant associations with all birth outcomes, including birth weight (β = 5907.30; 95% CI: −10,285.77 to −1528.86), birth length (β = −23.24; 95% CI, −45.71 to −0.78), birth head circumference (β = −20.66; 95% CI, −35.14 to −6.19), and birth chest circumference (β = −26.55; 95% CI, −44.41 to −8.69) (Table 4). Moreover, multiple linear regression analysis showed that VMR2 was correlated with birth weight (β = −4937.35; 95% CI, −8455.06 to −1419.64), birth length (β = −19.48; 95% CI, −37.39 to −1.58), birth head circumference (β = −18.75; 95% CI, −32.71 to −4.79), and birth chest circumference (β = −23.66; 95% CI, −38.44 to −8.87), whereas VMR1 and the vitamin D metabolites were not correlated with any neonatal anthropometric data (Table 4).

## 4. Discussion

The findings here show that most women we analyzed in this study were deficient in 25(OH)D, which is consistent with the findings of several previous reports [32,33]. Several studies on pregnant women of different ethnicities have shown insufficient vitamin D concentrations, as found in our study [12,39,40]. These studies have reported that 25(OH)D concentrations were not correlated with neonatal anthropometric measures. In an attempt to determine better parameters showing the associations between vitamin D and health outcomes, we found the ratio of 24,25(OH)_2_D to 25(OH)D to be a better marker for vitamin D sufficiency and bone health than 25(OH)D concentrations in older adults [41]. Therefore, the ratios of vitamin D metabolites may be more suitable than 25(OH)D alone for evaluating associations of vitamin D status with certain health outcomes. In fact, a report has shown an association between hip fracture risk and VMR1 in adults [28]. However, our data did not show any correlations between neonatal anthropometric data and VMR1. As 5-hydroxyvitamin D-24-hydroxylase (CYP24A1) converted 25(OH)D_3_ into 24,25(OH)_2_D_3_, VMR1 is considered to reflect the activity of VDR because it regulates CYP24A1 expression [28]. However, CYP24A1 expression is regulated by not only VDR but also parathyroid hormone and fibroblast growth factor 23 [42,43]. Both hormones are lower in fetuses than in adults [44,45,46]. Therefore, owing to the differences in these hormones, VMR1 may not be a robust indicator of VDR activity in neonates. This finding indicates that VMR1 is not a good indicator of bone metabolism in the neonatal period.

By contrast, this study showed that maternal VMR2 was significantly associated with anthropometric outcomes at birth, suggesting that VMR2 can be used as a predictor of these neonatal outcomes in pregnant women with low vitamin D status. In this study, VMR1 showed a strong correlation with 25(OH)D concentrations, suggesting that the production of 24,25(OH)_2_D_3_ depends on 25(OH)D concentrations. However, VMR2 only showed a weak correlation with 25(OH)D concentrations, suggesting that VMR2 could be an independent predictor of certain health outcomes. Regulatory factors other than 25(OH)D concentrations could be involved in the production of 3-epi-25(OH)D_3_.

Our results indicated that epimers were metabolites with potential correlations with neonatal anthropometric data in pregnant women who had low 25(OH)D concentrations. Epimers are increased in the pregnancy and neonatal periods [47,48,49]. Researchers have investigated pregnant women with low 25(OH)D concentrations and shown that 3-epi-25(OH)D_3_ was more correlated with health outcomes, such as cardiovascular complications of diabetes [31,50]. Epimers differ in their configurations around a single site of carbon atom (in this case, C-3α- vs. C-3β-hydroxy); otherwise, they have the identical chemical structures [51]. The epimerization of 25(OH)D_3_ at the C3 location is formed by 25(OH)D_3_-3-epimerase in the endoplasmic reticulum of liver, bone, and skin cells [52]. Epimerization may also be related to liver immaturity because infants show higher concentrations of epimers than adults [47]. Moreover, higher concentrations of epimers are detected in pregnant women than in non-pregnant women [48]. Studies have attempted to determine the physiological role of higher levels of C-3 epimers. In vitro experiments have shown that 3-epi-1,25(OH)_2_D_3_, which is metabolized from 3-epi-25(OH)D_3_ and is the most active form of the three 3-C epimers, showed nearly 10% transcription of the osteocalcin gene in human osteoblast cells compared with 1,25(OH)_2_D_3_. Moreover, the antiproliferative and differentiation-inducing activities of 3-epi-1,25(OH)_2_D_3_ were observed in 29% and 9.4% of 1,25(OH)_2_D_3_ forms, respectively [53]. Therefore, the transcriptional activity and biological effects of 3-epi-1,25(OH)_2_D_3_ are limited compared with those of its nonepimeric form. A future study focusing on the biological effects of epimers on fetal development is required.

Moreover, vitamin D metabolism may vary based on age; specific markers may be required in the neonatal period to consider vitamin D metabolism. In the future, an analysis of individuals of different ages, such as adults and/or older individuals, will also be required to determine whether vitamin D metabolism depends on age.

This study had several strengths. First, vitamin D and its metabolites were measured with high accuracy owing to modern technology. Second, this study was based on well-characterized, homogeneous Japanese pregnant women. Additionally, the covariates were well-adjusted because the data were based on a well-designed prospective cohort in Japan.

However, this study also had several limitations. First, we only measured maternal circulating blood vitamin D levels, not vitamin D levels in the uterus or placenta, which may be more relevant to neonatal growth. However, the blood of pregnant women may reflect the status of the interaction between mothers and fetuses. Moreover, blood sampling is more practical than placental or amniotic fluid sampling from the uterus during pregnancy. Second, this study was a complete case analysis, which could have caused bias due to potential differences among women with and without complete covariate data available.

In future research, our findings should be confirmed with another cohort comprising other ethnicities with low vitamin D levels. Clarifying the importance of maternal VMR2 as an independent indicator of birth outcomes will extend our knowledge and on the importance of vitamin D in the prenatal period.

## 5. Conclusions

In conclusion, VMR2 in Japanese pregnant women with vitamin D insufficiency is associated with neonatal anthropometric data. VMR2 during pregnancy appears to be an independent marker of 25(OH)D, which is important in predicting birth outcomes. Our results suggest that VMR2 should be determined when studying birth outcomes. Measuring and considering vitamin D metabolites may help interpret conflicting results regarding the associations between vitamin D status in pregnant women and birth outcomes.

## Figures and Tables

**Figure 1 nutrients-14-02201-f001:**
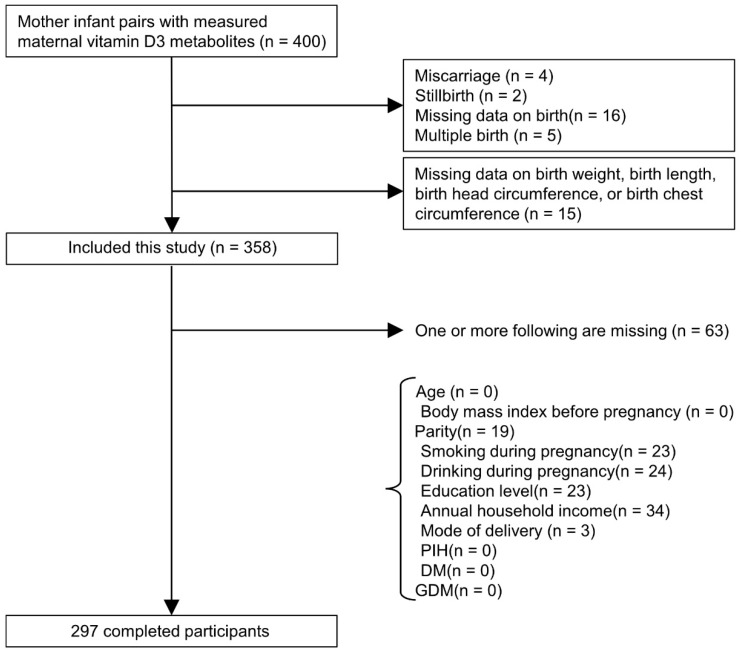
The flow chart of the participants included in this study.

**Figure 2 nutrients-14-02201-f002:**
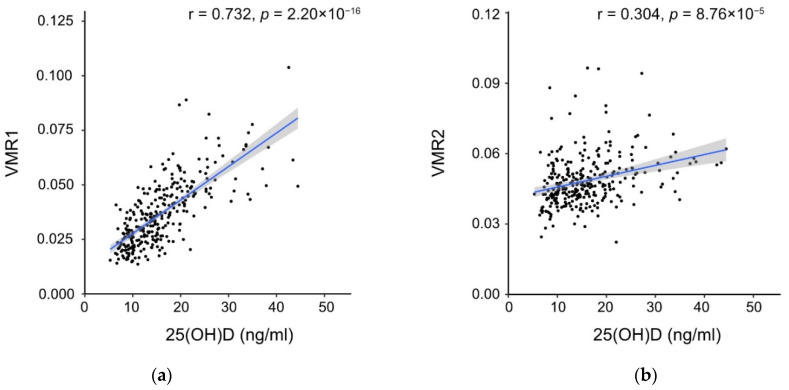
Scatter plots of VMR1 and 25(OH)D (**a**) and VMR2 and 25(OH)D (**b**). Scatter plots show that VMR1 has a strong correlation with 25(OH)D_3_ (r = 0.732; *p* = 2.20 × 10^−16^), and VMR2 has a weak correlation with 25(OH)D (r = 0.304; *p* = 8.76 × 10^−5^). 25(OH)D, 25-hydroxyvitamin D; VMR, vitamin D metabolite ratio.

**Table 1 nutrients-14-02201-t001:** The characteristics of the mothers included in this study.

Characteristics		All (*N* = 297)	
		*N*	%
Age at delivery (years), mean (SD)		31.3 (4.8)	
	<25	24	8.1
	25–<30	85	28.6
	30–<35	108	36.4
	35≤	80	26.9
Education (years)			
	<10	8	2.7
	10–<13	199	67.0
	13–<17	88	29.6
	17≤	2	0.7
Parity			
	0	107	36.0
	1≤	190	64.0
BMI before pregnancy (kg/m^2^), mean (SD)		21.5 (3.3)	
	<18.5	43	14.5
	18.5–<25	221	74.4
	25≤	33	11.1
Annual household income (million Japanese Yen)			
	<2	15	5.1
	2–<4	97	32.7
	4–<6	107	36.0
	6–<8	48	16.2
	8–<10	19	6.4
	10≤	11	3.7
Smoking habits			
	Never	159	53.5
	Quit before pregnancy	93	31.3
	Smoking early pregnancy	45	15.2
Alcohol consumption			
	Never	111	37.4
	Quit before pregnancy	57	19.2
	Drink early pregnancy	129	43.4
Pregnancy-induced hypertension		9	3.0
Gestational diabetes mellitus		10	3.4
Diabetes Mellitus		10	3.4
Vitamin D insufficiency (25(OH)D < 20 ng/mL)		253	85.2

BMI, body mass index; 25(OH)D, 25 hydroxyvitamin D; SD, standard deviation.

**Table 2 nutrients-14-02201-t002:** The characteristics of the infants included in this study.

		All (*N* = 297)	
		*N*	%
Sex			
	Male	145	48.3
	Female	152	50.7
Mode of delivery			
	Vaginal	233	77.7
	Cesarian	64	21.3
Birth weight (g), mean (SD)	3028.5 (416.0)		
Birth length(cm), mean (SD)	48.6 (2.1)		
Birth head circumference(cm), mean (SD)	33.2 (1.4)		
Birth chest circumference(cm), mean (SD)	31.7 (1.7)		
Gestational week, mean (SD)	39.2 (1.5)		

SD, standard deviation.

**Table 3 nutrients-14-02201-t003:** Levels of Vitamin D metabolites in maternal blood.

	Mean	SD	Minimum	25 Percentiles	Median	75 Percentiles	Maximum
25(OH)D (ng/mL)	15.82	7.10	5.31	10.31	14.63	19.27	44.45
25(OH)D_2_(ng/mL)	0.51	0.52	0.14	0.34	0.43	0.56	8.18
25(OH)D_3_ (ng/mL)	15.31	7.09	4.97	9.89	13.82	18.42	44.05
24,25(OH)_2_D_3_ (ng/mL)	0.66	0.56	0.08	0.27	0.46	0.85	4.42
3-epi-25(OH)D_3_ (ng/mL)	0.79	046	0.17	0.48	0.66	0.96	2.76
VMR1 (×10^−2^)	3.66	1.50	1.37	2.49	3.46	4.59	10.38
VMR2 (×10^−2^)	4.84	1.08	2.22	4.20	4.67	5.34	9.65

25(OH)D, 25-hydroxyvitamin D; 25(OH)D_2_, 25-hydroxyvitamin D_2_; 25(OH)D_3_, 25-hydroxyvitamin D_3_; 24,25(OH)_2_D_3_, 24,25-dihyroxyvitamin D_3_; 3-epi-25(OH)D_3_, 3-epi-25-hydroxyvitamin D_3_; VMR, vitamin D metabolite ratio; SD, standard deviation.

**Table 4 nutrients-14-02201-t004:** Estimated differences (and 95% CIs) in birth anthropometrics per 1 ng/mL increase in vitamin D metabolites or 1-unit increase in VMR.

Birth Weight (g)								
Model		25(OH)D	25(OH)D_2_	25(OH)D_3_	24,25(OH)_2_D_3_	3-epi-25(OH)D_3_	VMR1	VMR2
Unadjusted	β	0.28	66.08	−0.08	−8.63	−49.42	−155.80	−5907.30
	(95% CI)	(−6.44, 6.99)	(−25.49, 157.65)	(−6.80, 6.65)	(−93.07, 75.82)	(−153.68, 54.85)	(−3357.23, 3045.53)	(−10,285.77, −1528.86)
	*p*-value	0.936	0.157	0.982	0.841	0.352	0.924	**0.008**
Adjusted	β	−0.21	34.21	−0.39	−3.81	−46.58	−113.99	−4937.35
	(95% CI)	(−5.61, 5.20)	(−39.89, 108.31)	(−5.80, 5.02)	(−71.79, 64.17)	(−130.41, 37.26)	(−2693.21, 2465.23)	(−8455.06, −1419.64)
	*p*-value	0.940	0.364	0.888	0.912	0.275	0.931	**0.006**
Birth length (cm)								
Model		25(OH)D	25(OH)D_2_	25(OH)D_3_	24,25(OH)_2_D_3_	3-epi-25(OH)D_3_	VMR1	VMR2
Unadjusted	β	0.01	0.24	0.01	0.09	−0.11	3.63	−23.24
	(95% CI)	(−0.03, 0.04)	(−0.23, 0.70)	(−0.03, 0.04)	(−0.34, 0.52)	(−153.68, 54.85)	(−12.71, 19.97)	(−45.71, −0.78)
	*p*-value	0.621	0.321	0.672	0.681	0.697	0.662	**0.043**
Adjusted	β	0.00	0.04	0.00	0.09	−0.13	3.87	−19.48
	(95% CI)	(−0.02, 0.03)	(−0.34, 0.42)	(−0.02, 0.03)	(−0.26, 0.43)	(−0.56, 0.29)	(−9.18, 16.92)	(−37.39, −1.58)
	*p*-value	0.808	0.834	0.820	0.614	0.533	0.560	**0.033**
Birth head circumference (cm)								
Model		25(OH)D	25(OH)D_2_	25(OH)D_3_	24,25(OH)_2_D_3_	3-epi-25(OH)D_3_	VMR1	VMR2
Unadjusted	β	−0.01	0.13	−0.01	−0.07	−0.29	−3.38	−20.66
	(95% CI)	(−0.03, 0.01)	(−0.17, 0.44)	(−0.03, 0.01)	(−0.35, 0.21)	(−0.63, 0.06)	(−13.97, 7.21)	(−35.14, −6.19)
	*p*-value	0.369	0.395	0.335	0.614	0.100	0.530	**0.005**
Adjusted	β	−0.01	0.05	−0.01	−0.05	−0.27	−2.64	−18.75
	(95% CI)	(−0.03, 0.01)	(−0.25, 0.34)	(−0.03, 0.01)	(−0.32, 0.22)	(−0.60, 0.06)	(−12.86, 7.58)	(−32.71, −4.79)
	*p*-value	0.348	0.742	0.335	0.724	0.109	0.612	**0.009**
Birth chest circumference (cm)								
Model		25(OH)D	25(OH)D_2_	25(OH)D_3_	24,25(OH)_2_D_3_	3-epi-25(OH)D_3_	VMR1	VMR2
Unadjusted	β	0.00	0.36	−0.01	−0.11	−0.30	−4.23	−26.55
	(95% CI)	(−0.03, 0.02)	(−0.01, 0.73)	(−0.03, 0.02)	(−0.46, 0.23)	(−0.73, 0.12)	(−17.31, 8.85)	(−44.41, −8.69)
	*p*-value	0.749	0.060	0.647	0.513	0.164	0.525	**0.004**
Adjusted	β	−0.01	0.22	−0.01	−0.10	−0.30	−4.21	−23.66
	(95% CI)	(−0.03, 0.02)	(−0.10, 0.53)	(−0.03, 0.01)	(−0.38, 0.19)	(−0.66, 0.05)	(−15.08, 6.66)	(−38.44, −8.87)
	*p*-value	0.563	0.174	0.498	0.502	0.090	0.445	**0.002**

Models were adjusted for maternal age, body mass index (pre), alcohol habit, smoking habit, household income, education, gestational age, sex, and parity. VMR, vitamin D metabolite ratio. *p*-values marked with bold indicate statistically significant *p*-values.

## Data Availability

Data are unsuitable for public deposition owing to ethical restrictions and the legal framework of Japan. The Act on the Protection of Personal Information (Act No. 57 of 30 May 2003, amendment on 9 September 2015) prohibits public deposition of data containing personal information. The Ethical Guidelines for Medical and Health Research Involving Human Subjects enforced by the Japan Ministry of Education, Culture, Sports, Science and Technology and the Ministry of Health, Labour and Welfare also restrict open sharing of epidemiological data. All inquiries about access to JECS data should be sent to the following e-mail address: jecs-en@nies.go.jp. The person responsible for handling enquiries sent to this e-mail address is Dr Shoji F. Nakayama, JECS Programme Office, National Institute for Environmental Studies.

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
