# Peer review of "Vitamin D Metabolite Ratio in Pregnant Women with Low Blood Vitamin D Concentrations Is Associated with Neonatal Anthropometric Data"

_nutrients, 2022, doi:10.3390/nu14112201_

Round 1

Reviewer 1 Report

In my opinion, this study might be interesting for readers, especially in the time when vit.D became a very modern metabolite; several studies have shown that it has a significant influence on several aspects of human metabolism.

The number of patients is sufficient for analyses that were done. I have no comments or suggestions for the authors.

Author Response

Thank you for reviewing our manuscript and providing valuable feedback. Please find below a point-by-point response to the reviewers’ comments, with the responses and revisions highlighted in red in the main manuscript.

Reviewer 1

In my opinion, this study might be interesting for readers, especially
in the time when vit.D became a very modern metabolite; several studies
have shown that it has a significant influence on several aspects of
human metabolism.
The number of patients is sufficient for analyses that were done. I have
no comments or suggestions for the authors.

Response:Thank you for the positive feedback.

Reviewer 2 Report

This study on maternal vitamin D and its metabolite status in regard with neonatal anthropometric measurements is of interest.

  • Methods:
    1. The authors need to provide details on how the blood was collected and the storage conditions considering the serum used for analysis was collected ~10 years ago. Was any validation undertaken to establish the stability of vitamin D concentrations as storage conditions/length of time are reported to affect the levels measured (PMID: 33964192).
    2. Were any women on dietary supplements as this could influence the levels.
  • Results and Discussion: The authors need to address the reported literature findings in context with the current study as stated below.
    1. Numerous studies report that low maternal vitamin D levels during pregnancy is associated with higher risk of small for gestational age, low birth weight and prematurity. In the current study this does not appear to be the case despite ~85% of women having insufficient vitamin D concentration.
    2. The range reported is very broad - was there any seasonal influence which has been shown to influence maternal levels of circulating.
    3. Despite huge range in vitamin D and its metabolites, the newborn anthropometric measurements show minimal variation.
    4. Was there any sex difference?

Round 2

Reviewer 2 Report

The authors have satisfactorily addressed the comments.

Author Response

Thank you for your review of our manuscript.